# Hybrid cellular membrane nanovesicles amplify macrophage immune responses against cancer recurrence and metastasis

Lang Rao[1,8], Lei Wu[2,8], Zhida Liu [3,8], Rui Tian[1,4], Guocan Yu[1], Zijian Zhou[1], Kuikun Yang[1], Hong-Gang Xiong[2], Anli Zhang [3], Guang-Tao Yu[2], Wenjing Sun[4], Han Xu[4], Jingya Guo[5], Andrew Li[6], Hongmin Chen[4✉], Zhi-Jun Sun [2✉], Yang-Xin Fu [3,7✉] & Xiaoyuan Chen [1✉]

Effectively activating macrophages against cancer is promising but challenging. In particular, cancer cells express CD47, a 'don't eat me' signal that interacts with signal regulatory protein alpha (SIRPα) on macrophages to prevent phagocytosis. Also, cancer cells secrete stimulating factors, which polarize tumor-associated macrophages from an antitumor M1 phenotype to a tumorigenic M2 phenotype. Here, we report that hybrid cell membrane nanovesicles (known as hNVs) displaying SIRPα variants with significantly increased affinity to CD47 and containing M2-to-M1 repolarization signals can disable both mechanisms. The hNVs block CD47-SIRPα signaling axis while promoting M2-to-M1 repolarization within tumor microenvironment, significantly preventing both local recurrence and distant metastasis in malignant melanoma models. Furthermore, by loading a stimulator of interferon genes (STING) agonist, hNVs lead to potent tumor inhibition in a poorly immunogenic triple negative breast cancer model. hNVs are safe, stable, drug loadable, and suitable for genetic editing. These properties, combined with the capabilities inherited from source cells, make hNVs an attractive immunotherapy.

[1] Laboratory of Molecular Imaging and Nanomedicine, National Institute of Biomedical Imaging and Bioengineering, National Institutes of Health, Bethesda, MD 20892, USA. [2] State Key Laboratory Breeding Base of Basic Science of Stomatology (Hubei-MOST), Key Laboratory of Oral Biomedicine Ministry of Education, School and Hospital of Stomatology, Wuhan University, 430079 Wuhan, China. [3] Department of Pathology, University of Texas Southwestern Medical Center, Dallas, TX 75390, USA. [4] Center for Molecular Imaging and Translational Medicine, School of Public Health, Xiamen University, 361102 Xiamen, China. [5] Chinese Academy of Sciences Key Laboratory of Infection and Immunity, Institute of Biophysics, Chinese Academy of Sciences, 100101 Beijing, China. [6] Department of Biomedical Engineering, Johns Hopkins University, Baltimore, MD 21205, USA. [7] Department of Immunology, University of Texas Southwestern Medical Center, Dallas, TX 75390, USA. [8] These authors contributed equally: Lang Rao, Lei Wu, Zhida Liu. ✉email: hchen@xmu.edu.cn; sunzj@whu.edu.cn; yang-xin.fu@utsouthwestern.edu; chen9647@gmail.com

Cancer recurrence and metastasis account for over 90% of cancer-induced mortality[1,2]. Despite continuing improvements in surgical methods and other treatment modalities, residual microtumors and circulating tumor cells (CTCs) remain to be major stumbling blocks in cancer therapy[2,3]. Much effort have been invested in developing new and effective therapeutic approaches towards battling cancer recurrence and metastasis[4–6]. Among these approaches, immunotherapy for the purpose of activating the body's immune system against cancer has gained significant attention and stand ready to join traditional treatment modalities as an adjuvant treatment[7,8].

The innate immune system is a first line of defense in the body and macrophages play an indispensable role in its operation[9]. Effective activation of macrophages to 'eat' cancer cells holds great potential for cancer immunotherapy[10]. Unfortunately, cancer cells express CD47, which protects them from macrophage phagocytosis by sending a 'don't eat me' signal via the signal regulatory protein alpha (SIRPα) receptor[11,12]. Disrupting the CD47-SIRPα signaling axis has been explored as a potential immunotherapeutic strategy[13,14]. Our recent studies have suggested that CD47 blockade not only promotes the phagocytosis of cancer cells by macrophages but also boosts the antitumor T cell immunity[15], suggesting further potential in this emerging immunotherapeutic target. More than ten CD47 antagonists are currently being tested in clinical trials, while the objective response rate and clinical benefit rate of which require further improvements[16]. Also to note, systemic administration of CD47 antagonists can induce severe side effects including anemia and thrombocytopenia[8,16]. Efforts to address these concerns are critical in making anti-CD47 immunotherapy clinically feasible.

One mechanism that may be compromising the efficacy of CD47-blocking therapy is the immunosuppressive tumor microenvironment (TME) which is rich in signals that polarize tumor-associated macrophages (TAMs) towards a pro-tumorigenic M2 phenotype[17]. M2-type macrophages can recruit regulatory T cells (Tregs) and secret anti-inflammatory cytokines which counteract the activation of antitumor T cell immunity by CD47 blocking agents[8,18]. In this context, repolarization of TAMs from a pro-tumorigenic M2 phenotype to an antitumor M1 phenotype may restore efficacy to antitumor immunity of CD47 antagonists. In addition, the side effects and limited effectiveness may be caused by nonspecific binding of these CD47 blockades to normal tissues when systemically infused[5,16]. Thus, it would be ideal for CD47 blocking cancer immunotherapy to focus on the tumor site and avoid interactions in the immune milieu of the other sites.

Extracellular vesicles (EVs), including but not limited to exosomes and microvesicles, are lipid vesicles secreted by cells[19]. Recent evidence has suggested that EVs are involved in numerous physiological and pathological processes, and potentially have translational utility in immunotherapeutic agents for cancer treatment[20–22]. Due to their desirable safety characteristics and stability, using EVs as drug delivery vehicles attracted much attention[19,23,24]. In terms of production, however, the quantity of EVs secreted by cells is insufficient for use in drug delivery[19]. As a result, cellular nanovesicles (NVs), which can be produced by serial sonication and extrusion of cell membranes, have been utilized as alternatives to EVs[25–27]. More importantly, the NVs contain lipids and proteins of source cells and inherit multiple unique capabilities from the source cells[28]. For example, platelet-derived NVs (P-NVs) can interact with CTCs in the blood and bind to damaged vasculature and tissues[29–32]; M1 macrophage-derived NVs (M1-NVs) can repolarize TAMs to an M1-like phenotype[33,34]. Moreover, we have recently reported a facile method for fusing NVs derived from two different types of cells, resulting in hybrid NVs containing characteristics from both source cells[30].

In this study, we report a hybrid NVs (known as hNVs) that can amplify macrophage responses against cancer recurrence and metastasis after surgery. The hNVs consist of P-NVs, M1-NVs, and cancer cell-derived NVs overexpressing high-affinity SIRPα variants (SαV-C-NVs) (Fig. 1a). We show that the hNVs, which inherit the capabilities from source cells, can efficiently accumulate in surgical wound sites, interact with CTCs in the blood, repolarize TAMs towards M1 phenotype, and block the CD47-SIRPα interaction (Fig. 1b), thus accentuating macrophage phagocytosis of cancer cells, as well as potentiating antitumor T cell immunity, while reducing side effects induced by systemic infusion. In malignant melanoma models, we demonstrate that i.v. infusion of hNVs significantly prolong overall mouse survival by controlling both local recurrence and distant metastasis after surgery. Furthermore, by using a poorly immunogenic triple negative breast cancer model, we show that hNVs enhance the cytosolic delivery of a stimulator of interferon genes (STING) agonist and in turn, the agonist reprograms 'cold' tumors towards immunogenic states and improves the therapeutic efficacy of hNVs in a feedback manner. We anticipate the hNVs, which use only biocompatible cell membrane components, will provide additional insights on the development of safe and effective cancer immunotherapy strategy.

## Results

**Preparation and characterization of hNVs.** Briefly, the preparation of hNVs includes two steps: 1) obtaining engineered cells and the derived NVs, and 2) fusing single NVs to form the hNVs. SIRPα variant (SαV)[35] with 50,000-fold increased affinity to CD47 was first transduced onto B16F10 murine melanoma and 4T1 murine mammary carcinoma cell lines by lentivirus. We confirmed the expression of SαV on the cells by immunofluorescence imaging and flow cytometry (Fig. 1c, d and Supplementary Fig. 1). To extract these membranes, the intracellular content was removed by a combination treatment of hypotonic lysis, mechanical disruption, and gradient centrifugation. Subsequently, SαV-C-NVs were prepared by serial sonication and extrusion of the membranes through nanopores on a mini extruder. We also obtained M1-type macrophages by treating bone marrow-derived macrophages (BMDMs) with lipopolysaccharide (LPS) and prepared the M1-NVs with a similar protocol. In addition, LPS detection kit was used to verify the clearance of LPS in M1-NVs (Supplementary Fig. 2). Quantitative real-time polymerase chain reaction (qPCR) analysis demonstrated that M1-type TAMs and M1-NVs contained higher gene expression levels of pro-inflammatory markers (i.e., *Cd86*, *Il6*, *Tnf*, and *Inos*) when compared with nonpolarized M0 macrophages and the derived M0-NVs, respectively (Fig. 1e). Meanwhile, we separated platelets from mouse blood samples and obtained the derived P-NVs using similar methods (Supplementary Fig. 3).

After that, the resulting SαV-C-NVs, M1-NVs, and P-NVs were mixed, sonicated and extruded through nanopores repeatedly to form hNVs. Furthermore, pull down assay was employed to verify the membrane fusion and purify the hNVs (Supplementary Fig. 4). Dynamic light scattering (DLS) analysis and transmission electron microscopy (TEM) visualization revealed that the hNVs are round lipid droplets with an average size of 100 nm (Fig. 1f, g). To further determine if different types of NVs were indeed fused, single NVs were labeled with different fluorescent dyes before fusion. When the hNVs were viewed under a confocal microscope, significant overlap of fluorescent signals was observed (Fig. 1h), suggesting successful fusion of different types of NVs. Western blot analysis further showed that the hNVs contained specific

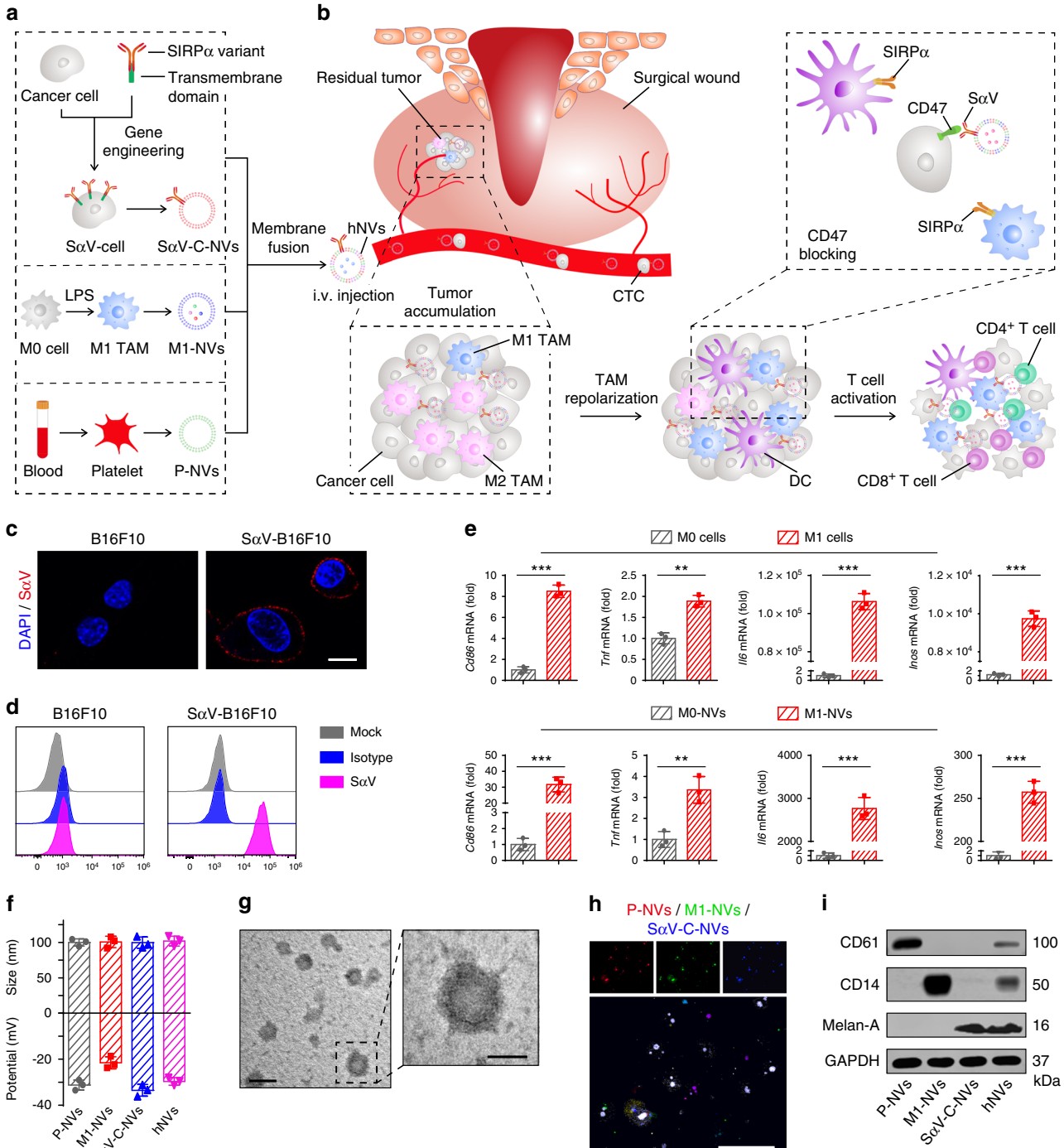

**Fig. 1 Schematic and characterization of hNVs. a** Schematic showing the hNVs consist of engineered SαV-C-NVs, M1-NVs, and P-NVs. **b** Schematic showing the hNVs efficiently interact with CTCs in the blood, accumulate in the post-surgical tumor bed, repolarize TAMs towards M1 phenotype, and block the CD47-SIRPα 'don't eat me' pathway, thus promoting macrophage phagocytosis of cancer cells, as well as boosting antitumor Tcell immunity. **c**, **d** Immunofluorescence imaging **c** and flow cytometry analysis **d** of SαV expression on original and engineered B16F10 cells. Scale bar, 10 μm. **e** Relative mRNA expression of *Cd86, Tnf, Il6, and Inos* in M0 cells, M1 cells, M0-NVs and M1-NVs. **f** Hydrodynamic size and zeta potential of P-NVs, M1-NVs, SαV-C-NVs, and hNVs measured by DLS. **g** TEM images of hNVs. Scale bar, 100 nm and 50 nm in the left and right panel, respectively. The samples were negatively stained with uranyl acetate. **h** Immunofluorescence images of hNVs. Scale bar, 10 μm. **i** Western blot analysis of specific protein CD61, CD14, and Melan-A in the samples of P-NVs, M1-NVs, SαV-C-NVs, and hNVs. All data are presented as mean ± S.D. ($n = 3$). Statistical significance was calculated via unpaired two-tailed t test. **$P < 0.01$; ***$P < 0.001$.

protein markers of individual NVs (Fig. 1i), including CD61, an important marker for platelet adhesion and activation; CD14, an endotoxin receptor on macrophages; and Melan-A, a melanoma tumor-associated antigen. Notably, the hNVs remained stable at least for two weeks in buffers (Supplementary Fig. 5) and demonstrated little to no cytotoxicity (Supplementary Fig. 6), reducing confounds in downstream in vitro and in vivo experiments.

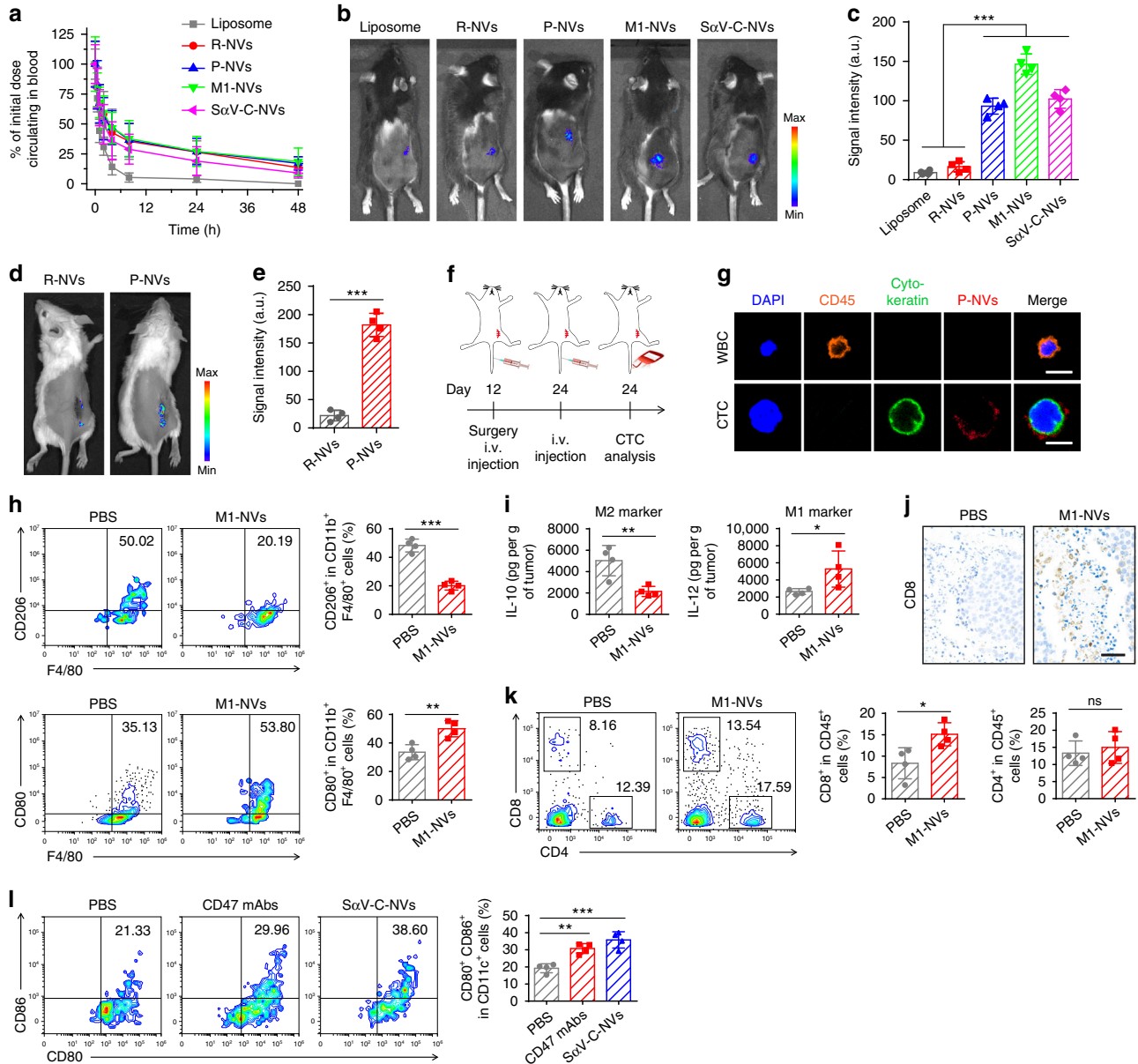

**Fig. 2 In vivo characterization of P-NVs, M1-NVs, and SαV-C-NVs. a**, In vivo pharmacokinetic curves of liposomes, R-NVs, P-NVs, M1-NVs, and SαV-C-NVs. **b, c** In vivo fluorescence imaging (**b**) and corresponding fluorescence intensities (**c**) of the mice 2 h after i.v. injection of equivalent dose of fluorescence-labeled liposomes and multiple NVs. **d, e** In vivo fluorescence imaging of the mice (**d**) and corresponding fluorescence intensities (**e**) 2 h after i.v. injection of equivalent dose of fluorescently labeled R-NVs and P-NVs. **f**, Schematic showing the CTC analysis schedule in a recurrence mouse model after incomplete surgery. **g** Immunofluorescence images of single WBC and CTC captured from blood samples. Scale bar, 10 μm. **h** Flow cytometric analysis of M2-like macrophages (CD206$^+$) and M1-like macrophages (CD80$^+$) in tumor gating on F4/80$^+$CD11b$^+$CD45$^+$ cells. **i** Secretion levels of IL-10 and IL-12 in different treatment groups. **j** Immunohistochemistry images of tumors showing CD8$^+$ T cells in different treatment groups. Scale bar, 50 μm. **k** Flow cytometric analysis of CD8$^+$ and CD4$^+$ T cells in tumor gating on CD45$^+$ cells. **l**, Flow cytometric analysis of CD80$^+$CD86$^+$ dendritic cells in tumor gating on CD45$^+$CD11c$^+$ cells. All data are presented as mean ± S.D. ($n = 4$). Statistical significance was calculated via ordinary one-way ANOVA with a Dunnett's test **c, l** or unpaired two-tailed $t$ test **e, h, i, k**. ns, no significance; *$P < 0.05$; **$P < 0.01$; ***$P < 0.001$.

**P-NVs targeting to post-surgery sites and interact with CTCs.** As with EVs, the phospholipid bilayer structure endows the NVs with long systemic circulation times[36]. The in vivo pharmacokinetics of multiple NVs after systemic infusion was assessed in a B16F10 mouse melanoma model. Liposomes and multiple NVs were first labeled with fluorescent dye and then intravenously (i.v.) injected into the mice. At indicated time points post-injection (p.i.), the blood was collected for fluorescence measurements. The NVs including red blood cell-derived NVs (R-NVs), P-NVs, M1-NVs, and SαV-C-NVs

showed superior retention in blood (Fig. 2a). In addition, the in vivo fluorescence imaging was performed at 2 h p.i., with tumor tissue and major organs collected for ex vivo imaging at 48 h p.i. We observed that P-NVs, M1-NVs, and SαV-C-NVs had better accumulation at tumor sites when compared with liposomes and R-NVs (Fig. 2b, c), relying on the specific targeting molecules on the NVs. Compared with liposomes, the NVs showed lower accumulation in the spleen and liver (Supplementary Fig. 7), which could reduce the side effects of the NVs on these organs[36].

Platelets are circulatory sentinels that respond to invasive microorganisms and vascular damage[4]. Recent studies have demonstrated that P-NVs inherit certain unique capabilities from the source platelets[29]. In a 4T1 mammary carcinoma spontaneous metastasis mouse model, we tested the capacity of P-NVs to accumulate in damaged tissues. After incomplete removal of tumor tissues, fluorescently labeled R-NVs and P-NVs were i.v. injected into the mice and in vivo fluorescence imaging was carried out at 2 h p.i. We observed that P-NVs were enriched in the injured tissue; whereas for the R-NVs, insignificant fluorescence signal was detected (Fig. 2d, e), suggesting that P-NVs inherited the damaged tissue targeting capability from platelets.

CTCs are malignant cells shed from solid tumors into the circulation system[37,38]. The recognition and interaction between platelets and CTCs are actively studied due to their key roles in cancer metastasis[4,29,39,40]. By using the 4T1 tumor metastasis model, we further investigated the interaction between P-NVs and CTCs. The mice were i.v. injected with fluorescently labeled P-NVs 24 days after tumor inoculation and the blood samples were collected from the mice at 2 h p.i. (Fig. 2f). The CTCs were separated by epithelial cell adhesion molecule antibody (anti-EpCAM)-modified immunomagnetic beads and identified by a typical immunofluorescence method, in which the combined information was used to delineate CTCs (CD45$^-$/Cytokeratin$^+$) from white blood cells (WBCs) (CD45$^+$/Cytokeratin$^-$)[30]. Remarkably, significant colocalization of P-NVs and CTCs was observed (Fig. 2g), demonstrating that the P-NVs could effectively bind to CTCs in vivo.

**M1-NVs and SαV-C-NVs induce potent immune responses**. M1 macrophage-derived microvesicles and exosomes, which contain mRNAs and miRNAs, can propagate pro-inflammatory signals and establish an immunostimulatory microenvironment[41]. To determine whether M1-NVs can also polarize M2 macrophages to M1 ones, we measured the mRNA levels in M2 macrophages after the M1-NVs treatment. Compared with untreated and M0-NVs groups, the M1-NVs showed lower levels of M2 markers (i.e., *Il4*, *Il10*, and *Fizz-1*) and higher levels of M1 markers (i.e., *Inos*, *Tnf*, and *Il6*) (Supplementary Fig. 8a, b). Cytokine measurement and immunofluorescence imaging further confirmed that M1-NVs could efficiently polarize M2 TAMs towards M1 ones (Supplementary Fig. 8c–f). Subsequently, the B16F10 mouse model was used to test the effects of M1-NVs on the TAM repolarization in vivo. Flow cytometry analysis demonstrated a decrease of M2-type macrophages (CD206$^+$CD11b$^+$F4/80$^+$) and an increase of M1 ones (CD80$^+$CD11b$^+$F4/80$^+$) within the tumor (Fig. 2h). In addition, the effects of M1-NVs concentration on macrophage polarization were tested (Supplementary Fig. 9). This polarization was verified by decreased level of IL-10 (M2 marker) and increased level of IL-12 (M1 marker) within the tumor (Fig. 2i). Notably, the M1-NVs treatment induced a significant increase in tumor-infiltrating T cells, especially CD8$^+$ T cells, within the TME (Fig. 2j, k), effectively improving the antitumor effects.

Disrupting the CD47-SIRPα signaling axis has been explored to be a promising immunotherapeutic strategy[42]. Multiple CD47 or SIRPα blockades, including anti-CD47 monoclonal antibodies (CD47 mAbs) and SIRPα-Fc fusion proteins, have shown promising antitumor efficacy in both preclinical models and clinical trials[8]. Given the weak interaction between native CD47 and SIRPα, SIRPα variants with improved affinity to CD47 were developed[39]. Unexpectedly, SIRPα variants are insufficient in inducing macrophage phagocytosis as single agents, but could serve as adjuvants to specific antibodies that opsonize tumor cells for destruction[35]. Interestingly, recent studies have suggested that

engineered exosomes could effectively activate macrophages to 'eat' cancer cells[43]. To investigate whether the SαV-C-NVs could stimulate macrophage phagocytosis, B16F10 cells were treated with SαV-C-NVs and then co-cultured with bone marrow-derived macrophages (BMDMs). Confocal imaging showed that CD47 blockade by SαV-C-NVs significantly increased the phagocytosis of cancer cells by BMDMs in a dose-dependent manner (Supplementary Fig. 10). Furthermore, the in vivo effects of SαV-C-NVs on the stimulation of macrophage phagocytosis were tested in a B16F10 mouse model. An increase in CD11c$^+$ dendritic cells was observed, and these cells exhibited higher expression of CD80 and CD86 (Fig. 2l), indicating their maturation status. By using the B16F10 mouse model, we compared the in vivo effects of different concentrations of SαV-C-NVs and CD47 mAbs on the inhibition of tumor growth (Supplementary Fig. 11).

**hNVs inhibit post-surgery recurrence of B16F10 tumors**. After confirming the unique capabilities of individual NVs, we further investigated the in vivo performance of hNVs in the treatment of tumor recurrence after surgery. The B16F10 incomplete-tumor-resection mouse model was used to imitate post-surgical tumor recurrence (Fig. 3a). After incomplete removal of tumor tissues by surgery, the mice were i.v. injected with three doses of phosphate buffered saline (PBS), P-NVs, M1-NVs, SαV-C-NVs, CD47 mAbs or hNVs every other day. Tumor growth was recorded by bioluminescence signal from luciferase-tagged cancer cells (Fig. 3b). The SαV-C-NVs and CD47 mAbs showed limited therapeutic effects on antitumor recurrence; in contrast, after they were integrated with P-NVs and M1-NVs, the resulting hNVs significantly reduced tumor recurrence as four out of six mice had no detectable tumor (Fig. 3c, d). As a result of suppression of tumor recurrence, the survival rate of the mice group was improved to about 66% after 60 days for the hNVs group (Fig. 3e). Notably, due to the synergistic effect of the surgical site and CTC targeting by P-NVs, the repolarization of TAMs towards an M1 phenotype by M1-NVs, and the blockade of CD47-SIRPα interaction by SαV-C-NVs, the hNVs showed even better antitumor effects than the cocktail therapy (simple mixture of three NVs) strategy (Supplementary Fig. 12). Meanwhile, benefiting from the tumor antigen specificity, B16-SαV-NVs showed better antitumor effects on B16F10 tumor models than 4T1-SαV-NVs (Supplementary Fig. 13). In addition, we also demonstrated that systemic administration of hNVs did not cause severe side effects to mice by body weight monitoring, serum biochemistry, complete blood and cytokine level test, and histology examination (Supplementary Figs. 14–16).

Residual tumors were collected and analyzed on 5th day after the administration. Flow cytometry results demonstrated an improved polarization of TAMs towards an M1 phenotype after the hNVs treatment (Fig. 4a). Compared with M1-NVs or SαV-C-NVs, the hNVs treatment markedly increased CD8$^+$ T cells within the TME (Fig. 4b), suggesting that the combination of CD47 blockade with M2-to-M1 repolarization effectively enhanced antitumor immunity. Immunohistochemistry and immunofluorescence analysis further confirmed significant increases in CD8$^+$ T cells and macrophages in the tumor (Fig. 4c, d). In addition, secretion of cytokines including IFN-γ, TNF-α, IL-10 and IL-12 suggested effective activation of innate and adaptive immune responses by hNVs (Fig. 4e)[44]. Notably, by using depleting antibodies against CD4$^+$ or CD8$^+$ T cells, we further demonstrated that the antitumor effects of hNVs are mainly dependent on CD8$^+$ T cells (Supplementary Fig. 17). Combining the capability of P-NVs to target sites of surgical resection, the repolarization of TAMs towards an M1 phenotype, and the blockade of CD47-SIRPα

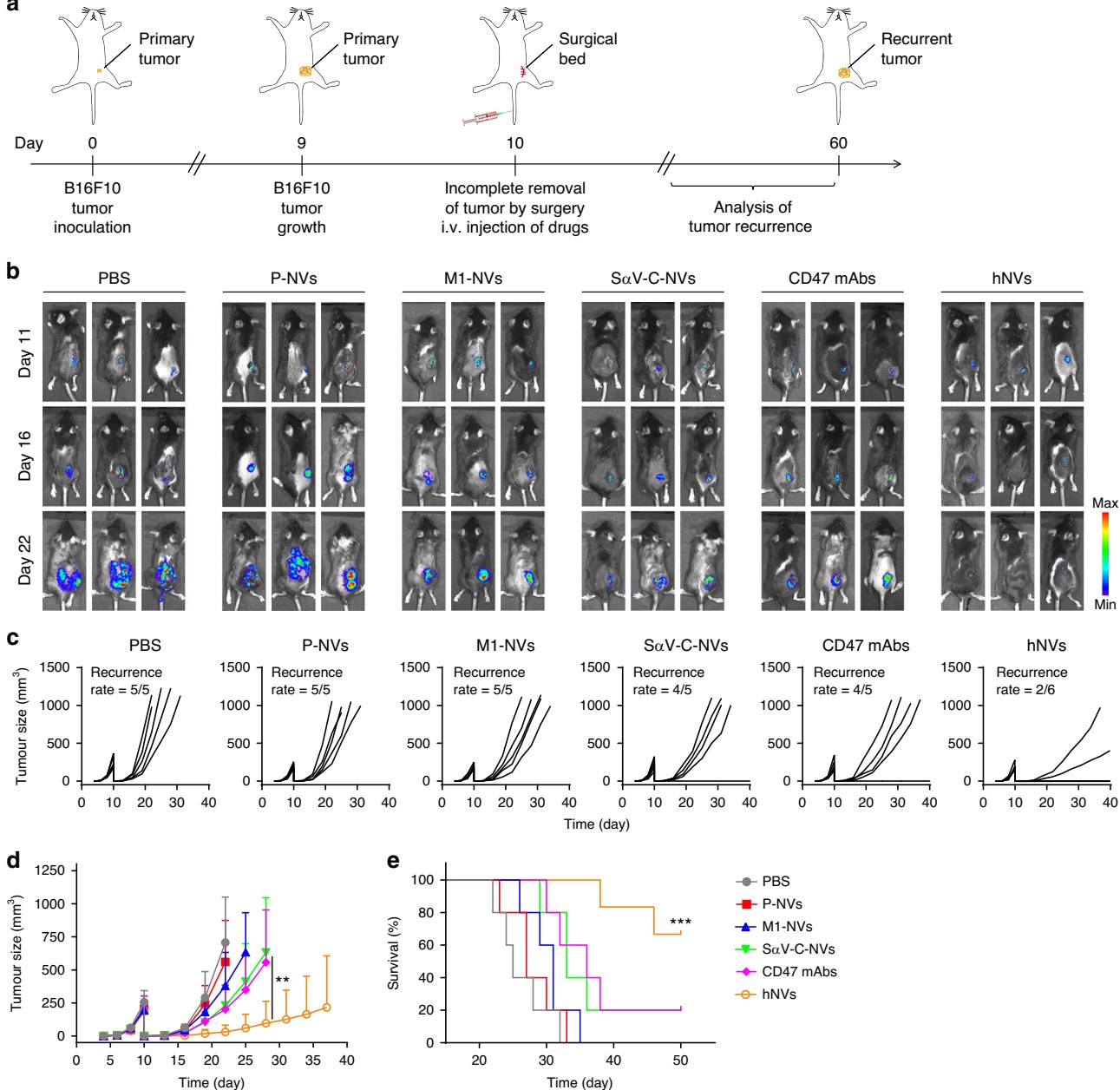

**Fig. 3 hNVs inhibit post-surgery recurrence of B16F10 tumors. a** Schematic showing the treatment schedule in a recurrence mouse model after incomplete surgery. **b** In vivo bioluminescence imaging of B16F10 recurrence in different treatment groups. **c, d** Individual **c** and average **d** tumor growth kinetics in different groups. Growth curves were stopped when the first mouse of the corresponding group died. **e** Survival corresponding to the tumor size of mice after different treatments as indicated. All data are presented as mean ± S.D. (n = 6 for the hNVs-treated group, n = 5 for the other groups). Statistical significance was calculated via 2way ANOVA with a Tukey's test **d** or log-rank (Mantel–Cox test) **e**. **P < 0.01; ***P < 0.001.

interaction, we have demonstrated the hNVs can effectively promote macrophage phagocytosis of cancer cells, as well as boost antitumor T cell immunity within the TME.

**hNVs inhibit post-surgery metastasis of B16F10 tumors.** Recent reports have suggested that surgery can promote cancer metastasis by releasing CTCs into the circulation system[2,45]. Thus eliminating the CTCs may help prevent cancer metastasis. To test the potential of hNVs in the elimination of CTCs, a metastasis model was developed by i.v. injection of mice with luciferase-tagged B16F10 cells after complete removal of tumor by surgery, thereby imitating the shedding of CTCs from the primary tumor into the systemic circulation (Fig. 5a). Immediately after surgery,

the mice were i.v. injected with three doses of PBS, CD47 mAbs or hNVs every other day. As assessed by the bioluminescence of cancer cells, CD47 mAbs treatment showed only mild anti-metastasis effects (Fig. 5b); in contrast, the hNVs therapy significantly reduced the lung metastasis (Fig. 5c, d), which can be attributed to the enhanced interaction with cancer cells in the circulatory system and the repolarized antitumor M1 type macrophages. Hematoxylin and eosin (H&E) and Ki-67 staining further confirmed effective inhibition of tumor cell proliferation in the lungs (Fig. 5e). Benefiting from effective control of tumor metastasis, half of the mice receiving hNVs remained alive after 50 days post the treatment (Fig. 5f), suggesting the great potential of hNVs in the treatment of cancer metastasis.

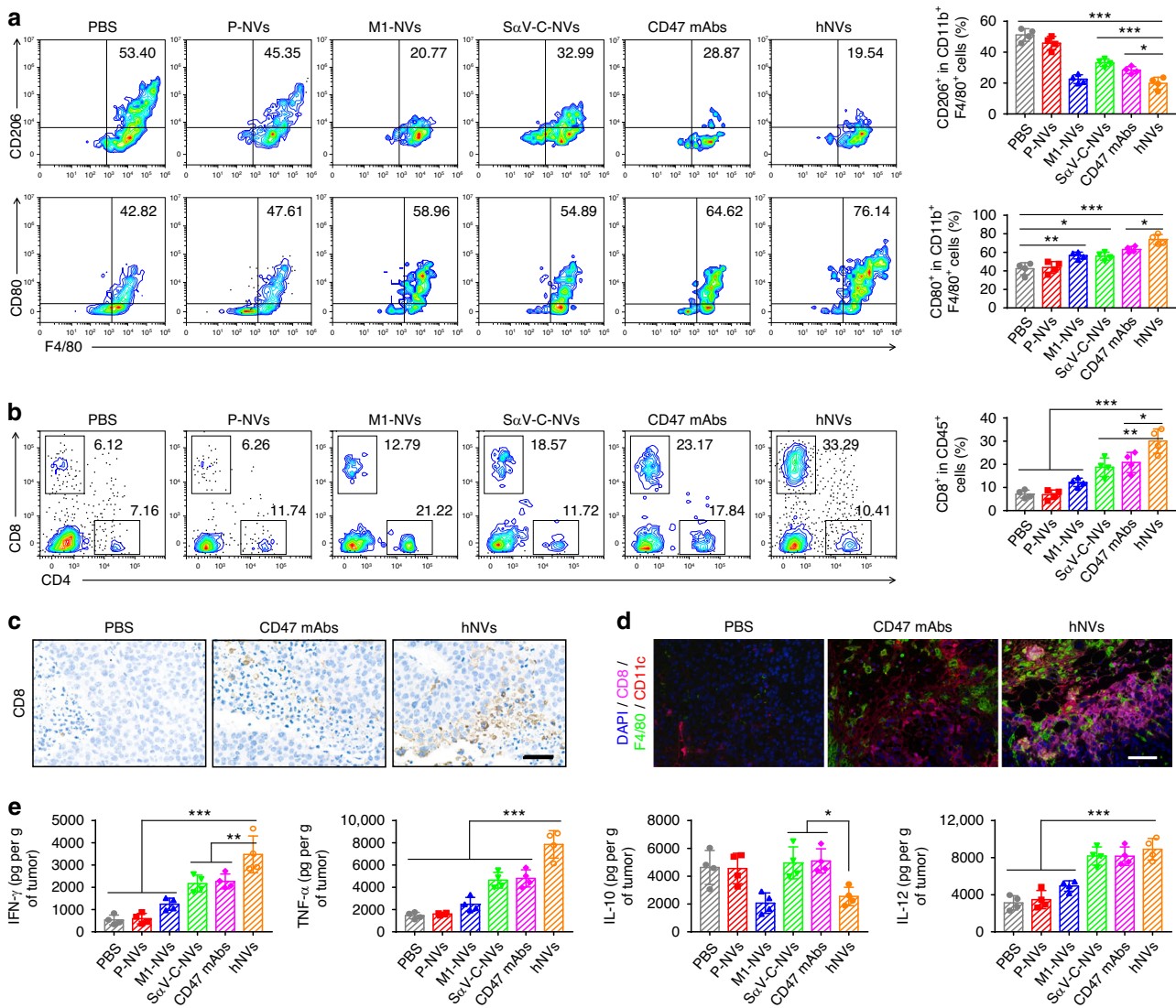

**Fig. 4 hNVs trigger potent antitumor immune response. a** Flow cytometric analysis of M2-like macrophages (CD206[+]) and M1-like macrophages (CD80[+]) in tumor gating on F4/80[+]CD11b[+]CD45[+] cells. **b** Flow cytometric analysis of CD8[+] and CD4[+] T cells in tumor gating on CD45[+] cells. **c** Immunohistochemistry images of tumors showing CD8[+] T cells in different treatment groups. Scale bar, 50 μm. **d** Multiplex immunohistochemistry images of tumors showing CD8[+] T cell and F4/80[+] macrophage infiltration in different treatment groups. Scale bar, 100 μm. **e** Cytokine levels in tumors from mice isolated 5 days after different treatments. All data are presented as mean ± S.D. ($n = 4$). Statistical significance was calculated via ordinary one-way ANOVA with a Tukey's test. *$P < 0.05$; **$P < 0.01$; ***$P < 0.001$.

**hNVs@cGAMP inhibit post-surgery recurrence and metastasis of 4T1 tumors.** To evaluate the effectiveness of hNVs in inhibiting another type of post-surgical cancer recurrence and metastasis, we conducted tests in a triple negative breast cancer 4T1 tumor model. After incomplete removal of the tumor by surgery, the mice were i.v. injected with three doses of hNVs every other day (Fig. 6a). However, beyond expectation, the hNVs showed limited antitumor effects in this model (Fig. 6b), which may be due to the anti-immunogenic TME created by 4T1 cancer cells.

Reprogramming 'cold' tumor towards immunogenic state, which reinvigorates antitumor T cell response, has recently attracted much attention[46]. STING is a cytosolic pattern recognition receptor that plays a critical role in spontaneous induction of antitumor T cell immunity[47,48]. The STING pathway is activated responding to abnormal DNA in the cytoplasm, which is detected by cyclic GMP-AMP synthase (cGAS), leading to the production of secondary messenger cyclic GMP-AMP

(cGAMP), the endogenous ligand for STING[49,50]. The key role of STING in tumor immunity has motivated many studies exploring cGAMP and related cyclic dinucleotide (CDN) agonists as therapeutic agents to boost antitumor immunity[9,51,52]. Although promising, the therapeutic efficacy of systemically delivered cGAMP is limited by the presence of several biological barriers, including rapid immune clearance, poor cell targeting and inefficient delivery to the cytoplasm where STING is located[53,54].

Unlike liposomes and other synthetic nanocarriers, EVs and NVs contain membrane-anchored and transmembrane proteins that can promote endocytosis and thus cytosolic delivery[23,24]. Inspired by this, we loaded cGAMP into the hNVs by sonication for enhanced cancer immunotherapy (Supplementary Fig. 18). To test the cytosolic delivery by hNVs, cGAMP in free or hNVs form was first conjugated with fluorescein and then co-incubated with bone marrow-derived dendritic cells (BMDCs). Confocal imaging revealed that hNVs@cGAMP had a higher fluorescence signal than free cGAMP in the cytoplasm of BMDCs (Supplementary

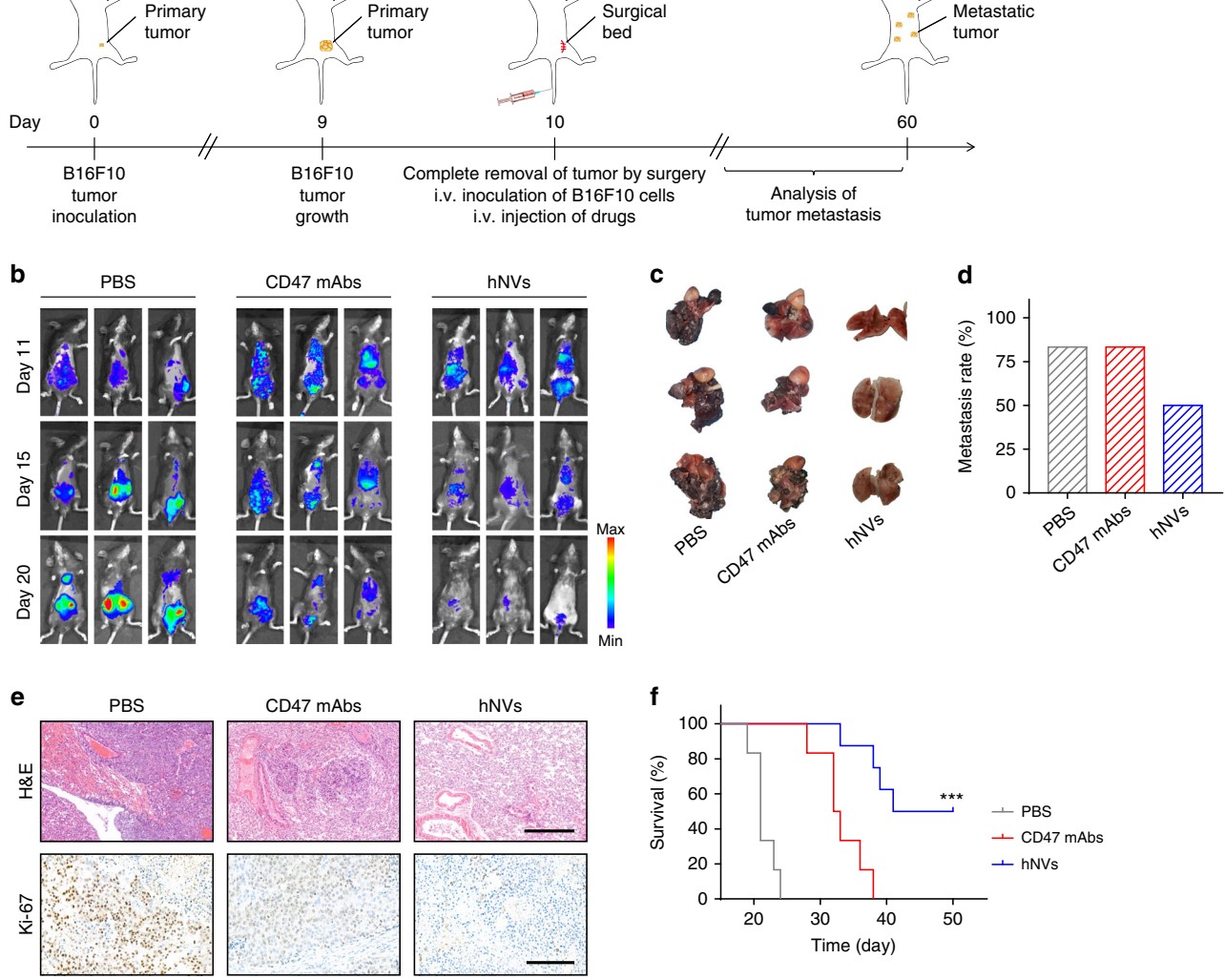

**Fig. 5 hNVs inhibit post-surgery metastasis of B16F10 tumors. a** Schematic showing the treatment schedule in a metastasis mouse model after complete surgery. **b** In vivo bioluminescence imaging of B16F10 metastasis in different treatment groups. **c** Lung photographs collected from the mice after indicated treatments. **d** Metastasis rates after indicated treatments. **e** H&E-stained and Ki-67-stained lung slices for different treatment groups. Scale bar, 500 μm and 200 μm in the H&E and Ki-67 slices, respectively. **f** Survival curves for different treatment groups. ($n = 8$ for the hNVs-treated group, $n = 6$ for the other groups). Statistical significance was calculated via log-rank (Mantel–Cox) test. ***$P < 0.001$.

Fig. 19a), suggesting hNVs efficiently improved the cytosolic delivery of cGAMP. In addition, qPCR analysis further confirmed that hNVs@cGAMP increased the gene expression of *Ifnb1*, *Cxcl9* and *Cxcl10* (Supplementary Fig. 19b), which play critical roles in the activation and recruitment of antitumor T cells. Furthermore, we demonstrated that hNVs entered into the target cells via endocytosis rather than plasma membrane fusion (Supplementary Fig. 20).

Subsequently, the post-surgery 4T1 tumor model was used to evaluate the antitumor effectiveness of hNVs. The mice were i.v. injected with three doses of PBS, hNVs, cGAMP, hNVs + cGAMP or hNVs@cGAMP every other day, immediately after surgery (Fig. 6a). Compared with hNVs or cGAMP alone, the hNVs@cGAMP exhibited effective control of tumor recurrence as four out of seven mice had no detectable tumor (Fig. 6b–d). Meanwhile, only very few metastatic foci were observed in the lungs of mice treated with hNVs@cGAMP (Fig. 6e–g), suggesting effective inhibition of lung metastasis. The mice receiving hNVs@cGAMP greatly benefited with regard to their survival, with more than 70% of them still alive on 60th day post the tumor inoculation (Fig. 6h). Furthermore, the increased expression of

*Ifnb1*, *Cxcl9*, and *Cxcl10* within tumors confirmed the enhanced delivery of cGAMP into cytoplasm by hNVs (Fig. 6i). Notably, hNVs@cGAMP exhibited even better antitumor effect than hNVs + cGAMP, which can be attributed to the fact that the cellular vesicles enhanced the cytosolic delivery of cGAMP (Supplementary Fig. 21) and in turn, cGAMP reprogrammed 'cold' tumors towards immunogenic states and promoted the therapeutic efficacy of cellular vesicles in a feedback manner.

## Discussion
In summary, we have developed a programmable cellular vesicle against cancer recurrence and metastasis after surgery. The P-NVs embedded in hNVs help to recognize and interact with CTCs in the blood and accumulate at the surgery site. M1-NVs and SaV-C-NVs could serve to repolarize TAMs towards M1 phenotype and block the CD47-SIRPα pathway, thus improving the phagocytosis of cancer cells by macrophages. In addition, CD47 blockade by SaV-C-NVs also stimulated the T cell-mediated devastation of cancer cells owing to the improved antigen presentation by macrophages and dendritic cells. In

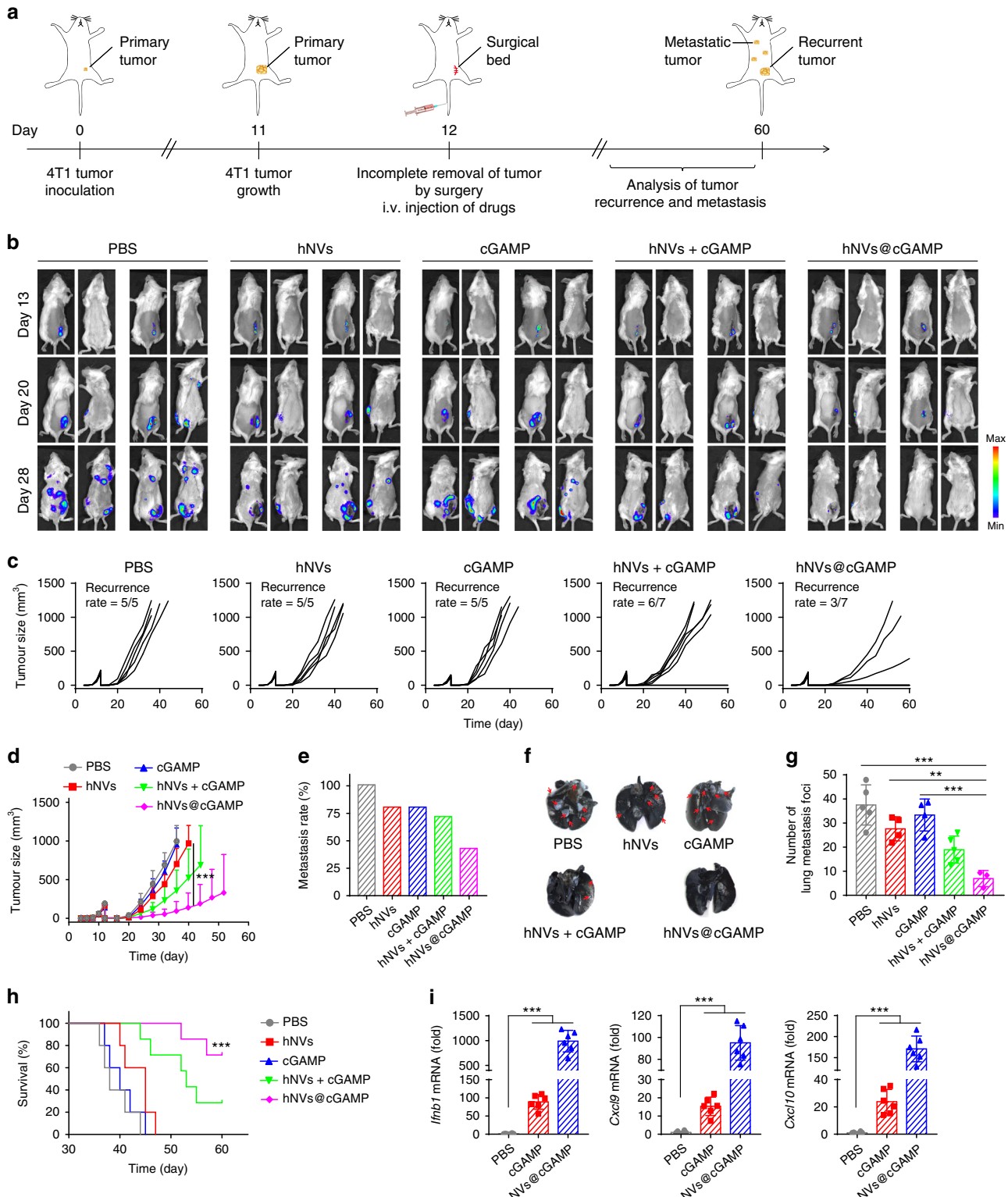

**Fig. 6 hNVs@cGAMP inhibit post-surgery recurrence and metastasis of 4T1 tumors. a** Schematic showing the treatment schedule in a recurrence and metastasis mouse model after incomplete surgery. **b** In vivo bioluminescence imaging of 4T1 recurrence and metastasis in different treatment groups. **c**, **d** Individual **c** and average **d** tumor growth kinetics in different groups. Growth curves were stopped when the first mouse died in the corresponding group. **e** Metastasis rates after indicated treatments. **f** Ink-stained lung photographs collected from the mice after different treatments. Red arrowheads indicate tumor foci in the lung. **g**, Numbers of lung metastatic foci after different treatments. **h** Survival curves for different treatment groups. **i** Relative mRNA expression of *Ifnb1*, *Cxcl9*, and *Cxcl10* in tumor 4 h after indicated treatments. Data are presented as mean ± S.D. (**d**, **h**, n = 7 for the groups treated with hNVs + cGAMP and hNVs@cGAMP, n = 5 for the other groups; **g**, n = 5 for the PBS group, n = 3 for the hNVs@cGAMP group, and n = 4 for the other groups; **i**, n = 6). Statistical significance was calculated via 2 way ANOVA with a Tukey's test **d** or ordinary one-way ANOVA with a Tukey's test **g**, **i** or log-rank (Mantel–Cox) test **h**. **$P < 0.01$; ***$P < 0.001$.

malignant melanoma mouse models, the hNVs exhibited significantly prolonged overall survival by controlling both local recurrence and distant metastasis after surgery. Although the hNVs alone showed mild antitumor effect in a poorly immunogenic triple negative breast carcinoma model, STING agonist, cGAMP-loaded hNVs significantly inhibited the post-surgical recurrence and metastasis by reprogramming 'cold' tumors towards immunogenic states. These promising results demonstrate the translational potential of this immunotherapy.

In realizing the clinical translation of hNVs, we should be vigilant about its in vivo toxicity. Much work is necessary for systematic assessment of potential short-term and long-term toxicity, but our small-scale pilot toxicity study, together with the safety data in several clinical studies regarding autologous EVs[19,55], could more or less reassure the safety concerns of our hNVs. Scalability is also a critical point that needs to be addressed for clinical relevance. Compared with the low quantity of EVs released from cells, the NVs are prepared by serial extrusion of cell membranes. By adopting large-scale purification and dispersion techniques widely used in biologics areas, reliable and high-throughput production of NVs can be envisioned.

Distinct from other synthetic nanocarriers, these NVs contain membrane-anchored and transmembrane proteins that can facilitate cytosolic delivery and certain membrane proteins on the NVs (e.g., CD47) that can protect the NVs from immune clearance[24,29]. Based on these findings, the hNVs platform can potentially serve as an robust platform for cytosolic delivery of nucleic acids and proteins. In addition, it is also known that membrane proteins are enriched in the lipid rafts of cell membranes. Based on this, NVs maybe also can be a platform for genetic editing for displaying functionalized membrane proteins, and can aid in correcting the spatial orientation and enhancing the activity of these biomolecules[56].

Another alluring feature of hNVs is that the cell membrane components can be individually customized, offering a high degree of freedom in programmable synthesis. While the current design employs platelet-derived, macrophage-derived, and cancer cell-derived NVs, it is conceivable that the hNVs platform can be generalized to many other types of NVs. Different types of NVs inherit different capabilities from source cells, such as bacterial NVs with immunity modulation ability and dendritic cell NVs capable of improving antigen presentation[57–59]. By generalizing this platform, programmable NVs can be developed for more robust cancer immunotherapy.

Looking towards the future, the use of NVs may open up an exciting field in personalized medicine. Tissues and cells can be collected from a patient after surgery and the derived NVs can be engineered before infusing them back into the same patient, which enables the maximization of immune tolerance to the NVs[60]. Abundant cell materials from blood and other tissue sources promise large-scale generation of NVs for post-surgical cancer immunotherapy. While the hNVs showed encouraging antitumor performance, the hNVs may contain suppressive factors that potentially halt antitumor immunity. Gene knockout of certain suppressive factors and combination with immune checkpoint inhibitors may further improve the antitumor effects of hNVs. Despite additional studies are necessary and further optimization can be done, the hNVs represent a significant technological advancement with the potential to expand the immunotherapeutic armamentarium.

## Methods
**Materials**. All chemicals were purchased from Sigma-Aldrich unless otherwise specified. CD47 mAbs were purchased from Bio-X cell (Clone, miap301). 2′3′-cGAMP VacciGrade were purchased from InvivoGen.

**Mice**. C57BL/6 mice (female, 6–10 weeks) and BALB/c mice (female, 6–10 weeks) were both purchased from Hunan Silaike Jinda Laboratory Animal Co. Ltd. (China). The animal study was approved by the Institutional Review Board of Wuhan University in accordance with the guidelines for the protection of animal subjects.

**Cells**. B16F10 murine melanoma, 4T1 murine mammary carcinoma, and 293T human embryonic kidney cell lines were all purchased from the American Type Culture Collection (ATCC). Luciferase-tagged B16F10 and 4T1 cells were established by transfection of B16F10 and 4T1 cells with vectors carrying luciferase and puromycin resistance gene. For construction of SIRPα variant-engineered cells, the cells were sorted and sub-cloned after being transduced by lentivirus expressing cell membrane bound SaV (SaV is an engineered high-affinity SIRPα variant fused with murine SIRPα transmembrane domain). The cells were cultured in 5% $CO_2$ and maintained in Dulbecco's modified Eagle's medium (DMEM) supplemented with 10% fetal bovine serum (FBS), 100 U/mL penicillin, and 100 μg/mL streptomycin (all from Invitrogen). Bone marrow-derived macrophages (BMDMs) were prepared following the steps blow. C57BL/6 or BALB/c mice were sacrificed and bone marrow cells were isolated from leg bones, maintained in RPMI medium supplemented with 10% FBS and 1% antibiotics, and differentiated with macrophage colony-stimulating factor (M-CSF) for 7 days.

**Preparation and characterization of single NVs**. For SaV-C-NVs, SaV-B16F10, or SaV-4T1 cancer cells were first suspended in hypotonic lysing buffer and disrupted by a Dounce homogenizer. The solution was treated with DNase and RNase (Invitrogen), and then centrifuged at $3200 \times g$ for 5 min. The supernatants were collected and further centrifuged at $20,000 \times g$ for 30 min, after which the supernatant was centrifuged again at $80,000 \times g$ for 1.5 h. The pellets were collected, washed with protease inhibitor tablet-mixed PBS for three times, sonicated for 5 min, and finally extruded through 400-nm, 200-nm, and 100-nm polycarbonate porous membranes on a mini extruder (Avanti Polar Lipids). In this work, autologous C-NVs were used for downstream in vitro and in vivo experiments.

For M1-NVs, M1 macrophages were first obtained by treating BMDMs with 100 ng/mL lipopolysaccharide (LPS; Sigma-Aldrich). The resulting M1 cells were resuspended in hypotonic lysing buffer, disrupted with the Dounce homogenizer, centrifuged at $3200 \times g$ for 5 min. The supernatants were pooled and centrifuged at $100,000 \times g$ for 2 h in a density gradient buffer formed by 10 and 50% OptiPrep layers using a LE-80K ultra-speed centrifuge (Beckman Coulter). The M1-NVs were collected from the interface of the layers and further centrifuged at $100,000 \times g$ for 2 h. Finally, the pelleted membranes were treated and characterized as described above. End-point chromogenic endotoxin test kit (BioEndo, China) was used to measure the residual LPS in the NVs.

For P-NVs, the whole blood was collected from mice and centrifuged at $100 \times g$ for 20 min. The supernatant was centrifuged again at $800 \times g$ for 20 min. The pelleted platelets were washed with PBS for three times, frozen at −80 °C, thawed at 25 °C, and centrifuged at $4000 \times g$ for 3 min. The pelleted membranes were treated and characterized as described above.

The preparation of multiple NVs was monitored by measuring the hydrodynamic diameter and zeta potential with a dynamic light scatter (DLS; Nano-Zen 3600, Malvern Instruments, UK). The morphologies of NVs were also observed by using a transmission electron microscopy (TEM; JEM-2010HT, JEOL, Japan). The TEM samples were prepared by contacting the droplet containing NVs with the copper grids for 60 s and then negatively staining with uranyl acetate for 30 s. The protein concentration of NVs was measured by using a Bradford reagent (Sigma-Aldrich).

**Preparation and characterization of hNVs**. After obtaining three single NVs, the hNVs were obtained according to a protocol we recently reported[30]. The P-NVs, M1-NVs and SaV-C-NVs were mixed (protein weight ratio of 1: 1: 4), sonicated for 5 min, and then extruded through 100-nm pores on the mini extruder. Anti-CD14 antibody-modified magnetic beads were prepared and pull-down assay was further used to purify the hNVs. After the purification, the protein contents in the sediments and supernatants of hNVs were measured by Bradford assay. The preparation of hNVs was monitored by DLS. The morphologies of hNVs were also observed by TEM. Immunofluorescence imaging was also used to determine whether different types of NVs were fused. Before the membrane fusion, P-NVs, M1-NVs, and SaV-C-NVs were labeled with 1,1′-dioctadecyl-3,3,3′,3′-tetra-methylindodicarbocyanine, 4-chlorobenzenesulfonate salt (DiD), 3,3′-dioctadecy-loxacarbocyanine perchlorate (DiO), and 1,1′-dioctadecyl-3,3,3′,3′-tetramethylindotricarbocyanine iodide (DiR) (all from Thermo Fisher Scientific), respectively. After the NV fusion, the hNVs were immobilized in glycerol and observed under a confocal laser scanning microscopy (CLSM; ZEISS LSM700).

**Immunofluorescence**. To confirm the SaV on the cells, parental and SaV-engineered B16F10 or 4T1 cells were plated into glass bottom dishes. After overnight culture, the cells were incubated with 1 μg/mL murine CD47-human IgG fusion proteins at 25 °C for 30 min. The cells were then stained with PE conjugated donkey anti-human IgG (Jackson ImmunoResearch) at 4 °C for 20 min. After being

stained with 4′,6-diamidino-2-phenylindole (DAPI), the cells were finally observed under the CLSM.

**Gene expression analysis.** For in vitro gene expression measurement, total RNA in the cells or NVs was purified by a RNeasy RNA Isolation Kit (Qiagen) and transcribed into cDNA by an iScript Reverse Transcription Supermix (Bio-Rad). qPCR was performed on a CFX96 Real-Time PCR detection system (Bio-Rad) with validated PrimePCR primers and SsoAdvanced Universal SYBR green Supermix. All primer sequences were provided in Supplementary Table 1.

For in vivo gene expression measurement, harvested tumors were placed immediately into RNAlater solution (Thermo Fisher Scientific) and stored at 4 °C overnight. On the next day, the samples were homogenized with zirconium beads on a BeadBug Homogenizer (Benchmark Scientific) and centrifuged to remove debris. The RNA was then treated and analyzed as described above.

**Western blotting.** The samples were denatured and loaded into 8–12% SDS-polyacrylamide gel. The proteins were then transferred onto polyvinylidene fluoride (PVDF) membranes, blocked with milk at 25 °C for 1 h, and incubated with primary antibodies: CD61 (ab119992; 1:1000 dilution), CD14 (ab221678; 1:1000 dilution), and Melan-A (ab210546; 1:1000 dilution; all from Abcam) at 4 °C overnight. The PVDF membranes were further incubated with horseradish peroxidase-conjugated secondary antibody (Thermo Fisher Scientific) and the blots were developed by using a West Pico PLUS Chemiluminescent Substrate kit (Thermo Fisher Scientific).

**In vivo pharmacokinetics and biodistribution.** $1 \times 10^6$ B16F10 cancer cells were subcutaneously (s.c.) injected into the right flank of C57BL/6 mice. On 8th day after the tumor inoculation, the mice were i.v. injected with 100 μL PBS containing 100 μg DiD-labeled liposomes, red blood cell-derived NVs (R-NVs), P-NVs, M1-NVs or SaV-C-NVs. At different time points, 20 μL blood were harvested from the tail veins. The harvested samples were diluted with 30 μL PBS and the fluorescence signal was measured by an IVIS imaging system (Perkin Elmer). At 2 h p.i., in vivo fluorescence imaging was performed using the same IVIS system. The mice hair was shaved with depilatory cream before in vivo imaging. At 48 h post the injection, tumor tissues and major organs were harvested for fluorescence imaging.

**In vivo evaluation of wound targeting and CTC interaction.** To measure the effects of wound targeting, $1 \times 10^6$ 4T1 cancer cells were s.c. injected into the right flank of BALB/c mice. On 12th day after the tumor inoculation, the tumor tissues were removed by surgery, remaining ~1% residual tumors to imitate the residual microtumors in the surgical bed. The wound was treated with an Autoclip wound closing system (Thermo Fisher Scientific). Immediately after the surgery, the mice were i.v. injected with 100 μL PBS containing 100 μg DiD-labeled R-NVs and P-NVs. At 2 h p.i., in vivo fluorescence imaging was performed using the IVIS system.

To further measure the effects of CTC interaction, the mice received another i.v. injection of DiD-labeled R-NVs and P-NVs on 24th day after the tumor inoculation. At 2 h p.i., the blood was collected from the mice for CTC analysis. 100 μg immunomagnetic bead technology (Thermo Fisher Scientific) were added into 1 mL of blood sample and mixed well at 25 °C for 60 min. The cells were then isolated with an external magnet. Captured CTCs were identified by immunofluorescence, in which the cells were stained with PE-anti-CD45, FITC-anti-Cytokeratin (all from B&D Biosciences), and DAPI before CLSM observation.

**Flow cytometry.** To confirm that SaV remained on the cells, parental and SaV expressing B16F10 or 4T1 cells were incubated with different concentrations of murine CD47-human IgG fusion proteins as indicated for 1 h at 4 °C. Then PE conjugated donkey anti-human IgG (Jackson ImmunoResearch) was used as a secondary antibody to stain the cells under 4 °C for 30 min. 7-AAD Viability Staining Solution was used to exclude the dead cells. Data were collected on a CytoFLEX flow cytometer (Beckman Coulter) and analyzed by using CytExpert (Beckman Coulter) software.

For flow cytometric analysis of tumor tissues, single cell suspensions were first prepared. The tumor masses were harvested from mice, cut into small pieces, and digested in RPMI medium containing 1 mg/mL collagenase D (Roche), 0.2 mg/mL DNases (Sigma-Aldrich), and 0.1 mg/mL hyaluronidase (Sigma-Aldrich) at 37 °C for 2 h, and were then filtered with 70-mm cell strainers (Becton and Dickinson). The cells were stained with fluorescence-labeled antibodies: CD11b (Biolegend, FITC, clone M1/70; 0.125 μg/test), CD11c (eBioscience, FITC, clone N418; 0.25 μg/test), CD206 (Biolegend, APC, clone C068C2; 0.25 μg/test), CD3 (eBioscience, FITC, clone 145-2C11; 0.5 μg/test), CD4 (eBioscience, FITC, clone GK1.5; 0.25 μg/test), CD45 (eBioscience, PerCP-Cyanine5.5, clone 30-F11; 0.125 μg/test), CD8 (Biolegend, APC, clone 53–6.7; 0.125 μg/test), CD80 (eBioscience, PE, clone 16-10A1; 0.06 μg/test), CD86 (eBioscience, APC, clone GL-1; 0.06 μg/test), F4/80 (Biolegend, PE-Cyanine7, clone BM8; 0.125 μg/test), Ly6C (eBioscience, APC, clone HK1.4; 0.125 μg/test), and Ly6G (Biolegend, PE, clone 1A8; 0.125 μg/test). The samples were analyzed on a CytoFLEX flow cytometer (Beckman) with CytExpert software (Beckman Coulter). Graphically account for all flow cytometric gating/sorting strategies were provided in Supplementary Fig. 22.

**Immunohistochemistry.** The tumor sections were deparaffinized, rehydrated, and treated with sodium citrate for antigen retrieval, followed by blocking endogenous peroxidase. The sections were incubated with CD8 antibody (1:1000 dilution; Cell Signaling Technology) and secondary antibodies, and then stained with ABC Kits (Vector Laboratories). The sections were scanned by using an Aperio ScanScope CS scanner (Vista).

**In vivo tumor models and treatments.** For the B16F10 tumor recurrence model, $1 \times 10^6$ luciferase-tagged B16F10 cancer cells were s.c. injected into the right flank of C57BL/6. On 10th day after the inoculation, the tumor tissues were removed by surgery, remaining ~1% residual tumors to imitate the residual microtumors in the surgical bed. After surgery, the mice received three doses of i.v. injection of PBS, P-NVs (50 μg per mouse), M1-NVs (50 μg per mouse), SaV-C-NVs (200 μg per mouse), CD47 mAbs (50 μg per mouse) or hNVs (300 μg per mouse) every other day. The in vivo fluorescence imaging was performed at indicated time points by using the IVIS system. Tumor volume was measured with a digital calliper.

For the B16F10 tumor metastasis model, $1 \times 10^6$ luciferase-tagged B16F10 cancer cells were s.c. injected into the right flank of C57BL/6. On 10th day after the inoculation, the tumor tissues were completely removed by surgery After surgery, the mice were i.v. injected with luciferase-tagged B16F10 cells, imitating the evasion of CTCs from the primary tumor and into the systemic circulation. The mice also received three doses of i.v. injection of PBS, CD47 mAbs (50 μg per mouse) or hNVs (300 μg per mouse) every other day. Metastatic burden was evaluated based on the bioluminescence of cancer cells. The in vivo fluorescence imaging was performed at indicated time points by using the IVIS system. The harvested lungs were fixed in 4% formalin, treated with paraffin, and sectioned at 4 μm. The lung slices were then stained with hematoxylin and eosin (H&E) or Ki-67 for further examination.

For the 4T1 tumor recurrence and metastasis model, $1 \times 10^6$ luciferase-tagged 4T1 cancer cells were s.c. injected into the right flank of BALB/c mice. On 12th day after the tumor inoculation, the tumor tissues were removed by surgery, remaining ~1% residual tumors. Immediately after the surgery, the mice received three doses of i.v. injection of PBS, hNVs, cGAMP, hNVs + cGAMP or hNVs@cGAMP (300 μg of hNVs and 36 μg of cGAMP per mouse) every other day. The tumor recurrence and metastasis burdens were monitored by the bioluminescence of cancer cells. The in vivo fluorescence imaging was performed at indicated time points by using the IVIS system. India ink was further used for the observation of lung metastases. The mice received an intratracheal injection of the ink (85 mL H₂O, 15 mL ink, two drops of ammonia water) before the sacrifice. The tumor tissues were then harvested and fixed with Fekete's solution. After 2–6 h, the tumor lesions were bleached whilst normal lung tissue remained the staining.

**Multispectral immunohistochemistry.** The formalin-fixed paraffin embedded tumor samples were used for Multicomplex immunofluorescence with Perkin Elmer Tyramide Plus (Opal) reagents according to the manufacturer's instructions. The paraffin sections were first deparaffinized, rehydrated, treated with AR buffer for antigen retrieval, covered with blocking buffer, and then were incubated with primary antibody, followed by secondary antibody. Sections were washed for three times in 0.02% Tris-buffered saline–Tween 20 (TBST), followed by signal generation with 100 μL Opal Fluorophore Working Solution, incubated at 25 °C for 10 min. Opal 570 Fluorophore, Opal 650 Fluorophore, and Opal 690 Fluorophore (all buffers, antibodies and dyes from Perkin Elmer) were applied to each antibody. Multispectral images were acquired by PerkinElmer Vectra platform at ×20 magnification. The following primary antibodies were used in this panel: CD8 (1:400 dilution), F4/80 (1:400 dilution), CD11c (1:400 dilution; all from Cell Signaling Technology), and DAPI.

**Cytokine detection.** Single cell suspensions were first prepared by harvesting tumor tissues on 5th day after last injection and homogenizing in cold PBS supplemented with digestive enzyme. The intratumor levels of IFN-γ, TNF-α, IL-10, and IL-12 were detected with ELISA kits (all from Invitrogen) according to the measurement manual.

**Statistical analysis.** All results are presented as mean ± standard deviation (S.D.). The unpaired two-tailed t test was used for two group comparisons and ordinary one-way (or 2way) ANOVA with a Tukey's test (or with a Dunnett's test) were used for multiple group comparisons. The log-rank (Mantel–Cox) test was used to determine the mouse survival benefit. All statistical analyses were performed with Prism 5.0 software (GraphPad).

**Reporting summary.** Further information on research design is available in the Nature Research Reporting Summary linked to this article.

## Data availability

All relevant data are available in the article, supplementary information, or from the corresponding author upon reasonable request. Source data are provided with this paper.

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

## Acknowledgements

This work was supported by the Intramural Research Program of the National Institute of Biomedical Imaging and Bioengineering (NIBIB), National Institutes of Health (NIH), the Cancer Prevention Research Institute of Texas (CPRIT) Grants (RR150072 and RP180725), the National Key Research and Development Program of China (2018YFA0107301), and the National Natural Science Foundation of China (81874131).

## Author contributions

L.R., L.W., Z.L., Y.-X.F., and X.C. conceived and designed the experiments. L.R., L.W., Z.L., R.T., G.Y., Z.Z., K.Y., H.-G.X., A.Z., G.-T.Y., W.S., H.X., and J.G. performed the experiments. L.R., L.W., Z.L., R.T., H.C., Z.-J.S., Y.-X.F., and X.C. analyzed the data. L.R., Z.L., R.T., G.Y., A.L., Y.-X.F., and X.C. wrote the manuscript. All authors have given approval to the final version of the manuscript.

## Competing interests

The authors declare no competing interests.
