## [Peer Review File · Nature Communications]

Reviewers' comments:

Reviewer #1 (Remarks to the Author): expertise in macrophages; CD47/SIRP axis

I apologize for the delay in returning my comments. I was dealing with the COVID-19 crisis.

In this paper, the authors examined the impact of cell-derived nanovesicles (NVs) on tumor growth and metastasis in mice. NVs were produced from B16 cells expressing a high-affinity variant of SIRP α , LPS-treated macrophages and platelets. NVs were then injected in mice bearing B16 cells or 4T1 cells. NVs were shown to have an apparent half-life of ~4 hours and to localize to tumors. They were well tolerated. iNVs, which contained all three types of NVs, had some antitumor efficacy against B16, but not 4T1. Addition of a STING agonist enabled some effect of the iNVs against 4T1.

While these data are of some interest, they are highly descriptive. It is not clear at all how the iNVs are mediating their effects. There is a lack of mechanistic insight. Moreover, there is a lack of key relevant controls.

Specific comments:

- 1) NVs from B16 lacking SIRP α or from non-LPS-treated macrophages, alone or in combination, should be used.
- 2) Also, only iNVs and no single component NVs were used in the 4T1 model. Why? What about the single component NVs with the STING agonist.
- 3) The authors need to verify that there is no residual LPS in their M1-NV prep.
- 4) A more detailed insight into the stability and specificity of tumor targeting should be provided.
- 5) Are the NVs internalized or fused with plasma membranes of target cells?

Reviewer #2 (Remarks to the Author): expertise in STING agonist in cancer immunotherapy

Designer cell membrane nanovesicles amplify macrophage immune responses against cancer recurrence and metastasis

Here, Lang et al. designed novel nanovesicles (NVs), made up from the fusion of platelet-derived, M1 macrophage-derived and genetically-engineered (overexpressing high affinity CD47 receptor) cancer cell-derived extracellular vesicles and utilized them for cancer immunotherapy upon comprehensive characterization. Importantly, fusion vesicles (or so-called iNVs) showed potent anti-tumor effect by abrogating cancer recurrence and metastasis in murine tumor models of melanoma and to a lesser extent in triple negative breast cancer model, for which incorporation of an adjuvant, like cGAMP, into the NVs was necessary in order to get the optimal anti-tumor effect due to low immunogenicity of this tumor model. Mechanisms regulating the anti-tumor effect of iNV immunotherapy involved M1 polarization of TAMs, enhanced macrophage phagocytosis due to blockade of CD47-SIRP α signaling on TAMs and subsequent T cell recruitment to tumors. Therefore, such specially-designed NVs may offer therapeutic applications for cancer.

Reviewer Comments:

1- As the authors also discussed, application of the anti-tumor NVs to the clinic is difficult, especially to produce clinical grade/GLP lot NVs. In the discussion, authors suggested that patient-derived NVs could be used to produce such anti-tumor NVs. However, have the authors examined the possibility that cancer patient-derived NVs may contain suppressive factors that can potentially halt anti-tumor immunity. Do they have any suggestion for this issue?

2- Starting from figure 3, authors compare therapeutic approaches using different types of NVs or anti-CD47 antibody treatment in murine tumor models. However, they compare 50 μ g of P-NV, M1-NV, anti-CD47 antibodies with 200 μ g of SaV-C-NV and 300 μ g of iNVs. Why do they use

different doses for each agent rather than directly comparing the same dose of each NVs at least? Can the authors explain rationale behind this strategy?

3- Authors obtain M1-NVs from LPS-stimulated BMDMs raising a potential LPS contamination issue in M1-NVs, which are systemically injected to animals. Have the authors measured serum cytokines, such as IL-6, TNF α or type I IFNs, after i.v. injection of these NVs or observed any adverse effects related to this issue?

4- NV treatment can enhance infiltration of not only CD8 but also CD4 T cells as observed in FACS plots in figures 2k and 4b into the TME. Have the authors examined those CD4 T cells? Are they Tregs or anti-tumor T cells? In other words, is the NV-mediated anti-tumor effect dependent on CD8 or CD4 T cells?

5- In this study, authors designed iNVs by using the same type of tumor cells for each tumor model (e.g. B16 F10 NVs for melanoma model and 4T1 NVs for breast cancer model). Have the authors tried the vice versa, for example, using B16 F10 NVs in breast cancer model in order to show the tumor antigen-specificity of this anti-tumor effect? Another way to prove this would be utilization of a tumor cell line expressing a model antigen, such as OVA, and measure antigen-specific T (or even B) cell responses after ex vivo splenocyte re-stimulation (or measuring OVA-specific serum antibodies).

Reviewer #3 (Remarks to the Author): expertise in membrane derived nanovesicles

In this article, Rao et al. fused engineered cancer cell nanovesicles, platelet nanovesicles, and M1 macrophage nanovesicles into a single platform for the prevention of tumor recurrence and metastasis in a surgical setting. The iNVs had clear efficacy in surgical models and is versatile enough for drugs to be easily incorporated inside. While the benefits of the platelet and macrophage component was clearly shown, the need and advantages for the cancer cell nanovesicle component was lacking. It is unclear as to why the authors decided to study the benefits of each individual component without a comparison to their actual formulation. Moreover, the evidence for the nanovesicle fusion is limited. Based on the results provided, it is unclear if the authors actually fused the three distinct components together or simply administered them as a cocktail formulation. Below are some detailed comments for the authors to further improve the manuscript.

1. While the roles of P-NVs and M1-NVs were clearly defined, the necessity for engineered C-NVs is unclear. Can the author elaborate on this design?
2. Throughout the manuscript, the authors referred to M0-NVs several times, but did not explain what they are. Are these NVs derived from unstimulated BMDMs? Please explain.
3. For the fusion membrane, the author chose a protein ratio of 1:1:4 for P-NVs:M1-NVs:SaV-C-NVs. Can the authors please explain and provide data as to how this ratio was determined? Why was it chosen as opposed to a simple 1:1:1?
4. While the fluorescence images in Figure 1h appears to show that there are some fusion NVs due to colocalization, the data itself is insufficient. The supporting WB analysis is a flawed design since there's no purification step involved in the fusion process. One can in theory get the same WB results if they just simply mix the three components at the same ratio. Please provide more supporting evidence that the final NVs are indeed a fusion of the three, particularly if the authors can show that it's fused at a 1:1:4 manner.
5. Cytotoxicity with a cancer cell line in Figure S4 is not a good representation of biocompatibility. The authors should repeat the experiment with a cell line of non-cancerous origin or with primary cells.
6. The schematic in Figure 2d was quite confusing. From the manuscript, it seems like these were two separate experiments. Can the authors revise it and try to make it clearer?

7. The mechanism of M1-M2 polarization is unclear. While there has been studies where extracellular vesicles can induce M2 polarization, that interaction is largely due to mRNAs/miRNAs that are naturally inside those vesicles. In this platform, those compounds should not exist since the cells were lysed and then the NVs are reformed. What is actually allowing M1-NVs to polarize M2 macrophages into the M1 phenotype?

8. The distinct dosages the authors chose for their therapeutic studies are quite odd, which can lead to an unfair comparison between the different groups. In the current scheme, the advantage of the fusion iNVs is abrogated. Arguably, the same therapeutic efficacy can be achieved with a cocktail administration. The authors should normalize the dosage to the same protein content in order for a fairer comparison.

9. In Figure 3c, why does the iNVs group have 6 mice whereas all the other groups have 5 mice? This seems like an unfair comparison. Similarly, this is observed in Figure 6c for iNVs + cGAMP and iNVs@cGAMP groups. The authors should keep consistently the same number of mice in each group for their studies.

Point-by-point response to review comments

Reviewer #1 (Remarks to the Author): expertise in macrophages; CD47/SIRP axis

I apologize for the delay in returning my comments. I was dealing with the COVID-19 crisis. In this paper, the authors examined the impact of cell-derived nanovesicles (NVs) on tumor growth and metastasis in mice. NVs were produced from B16 cells expressing a high-affinity variant of SIRPa, LPS-treated macrophages and platelets. NVs were then injected in mice bearing B16 cells or 4T1 cells. NVs were shown to have an apparent half-life of ~4 hours and to localize to tumors. They were well tolerated. iNVs, which contained all three types of NVs, had some antitumor efficacy against B16, but not 4T1. Addition of a STING agonist enabled some effect of the iNVs against 4T1. While these data are of some interest, they are highly descriptive. It is not clear at all how the iNVs are mediating their effects. There is a lack of mechanistic insight. Moreover, there is a lack of key relevant controls.

Response: Thank you very much for your insightful comments. According to your comments, we tried our best to revise the manuscript. More than 100 mice were involved in the revision to include more experimental controls. We also addressed the concerns related to the potential LPS contamination and the entry of NVs into target cells. We marked all the changes in red and point-by-point responses to your comments are listed below.

Specific comments:

1) NVs from B16 lacking SIRPa or from non-LPS-treated macrophages, alone or in combination, should be used.

Response: Per your suggestion, we added the NVs from B16 lacking SIRPa (C-NVs) and from non-LPS-treated macrophages (M0-NVs) as control and repeated the in vivo experiments based on B16F10 tumor model. As shown in **Figure S12 (see below)**, M0-NVs and C-NVs had mild antitumor effects on B16F10 model. More discussion and experiment details can be found in the revised manuscript and supplementary information.

Supplementary Figure S12. c, Tumor growth kinetics in different groups. d, Survival corresponding to the tumor size of mice after different treatments as indicated.

2) Also, only iNVs and no single component NVs were used in the 4T1 model. Why? What about the single component NVs with the STING agonist.

Response: Per your suggestion, we added single NVs loaded with cGAMP (i.e., P-NVs@cGAMP, M1-NVs@cGAMP, and SαV-NVs@cGAMP) as control and repeated the in vivo experiments 4T1 tumor model. Although the NVs could enhance the intracellular delivery of cGAMP (Figure S19), the antitumor effects of single NVs are limited (Figure S12). Therefore, compared with iNVs@cGAMP, single NVs loaded with cGAMP only showed mild antitumor effect on 4T1 tumor mice (Figure S21). More discussion and experiment details can be found in the revised manuscript and supplementary information.

Supplementary Figure S21. c, Tumor growth kinetics in different groups. d, Survival corresponding to the tumor size of mice after different treatments as indicated.

3) The authors need to verify that there is no residual LPS in their M1-NV prep.

Response: According to your suggestion, **End-Point Chromogenic Endotoxin Test Kit (BioEndo, China)** was used to measure the residual LPS in the NVs. As shown in Figure S2, the residual LPS in the iNVs was not detectable. The limit of detection of this LPS Kit is 0.01 – 0.1 EU/mL, thus **there are less than 0.01 EU/mL LPS remaining in the iNVs, meeting the requirements of in vivo experiments regarding LPS.**

In addition, we also tested the serum cytokine level after the iNVs treatment and **observed no significant differences in TNF-α and IL-6 (Figure S16)**, further confirming the biosafety of iNVs.

Supplementary Figure S2. Measurement of LPS contents in M0-NVs, M1-NVs and iNVs.

Supplementary Figure S16. Serum cytokine measurement. **a**, TNF- α : tumor necrosis factor- α . **b**, IL-6: interleukin-6.

4) A more detailed insight into the stability and specificity of tumor targeting should be provided.

Response: For P-NVs targeting to damaged tissues, in 2015, Zhang et al. demonstrated in detail that the targeting ability of P-NVs to damaged tissues is **dependent on cytokine and chemokine receptor proteins (Nature 2015, 526, 118-121)**. We also demonstrated that platelet membrane-coated nanoparticles could efficiently accumulate in the photothermal therapy-injured tumor tissues (Adv. Funct. Mater. 2017, 27, 1604774).

For P-NVs interacting with CTCs, the recognition and interaction between platelets and CTCs have been actively studied due to their key roles in cancer metastasis (Nat. Rev. Cancer 2011, 11, 123-134). In 2017, Gu et al. demonstrated that the interaction between platelet vesicles and CTCs is **dependent on P-selectin protein** and further used in situ activated platelet vesicles for anticancer metastasis application (Nat. Biomed. Eng. 2017, 1, 0011).

Considering that the mechanisms of P-NVs targeting to damaged tissues and interacting with CTCs have been well studied and these capabilities have also been widely used for biomedical applications, we didn't pay much attention on the mechanism insight, but focused on utilizing these unique properties of P-NVs for enhanced cancer immunotherapy.

In the revised work, we added more discussion related to tumor targeting, such as "Platelets are circulatory sentinels that respond to invasive microorganisms and vascular damage. Recent studies have demonstrated that P-NVs inherit certain unique capabilities from the source platelets" and "CTCs are malignant cells shed from solid tumors into the circulation system. The recognition and interaction between platelets and CTCs are actively studied due to their key roles in cancer metastasis".

5) Are the NVs internalized or fused with plasma membranes of target cells?

Response: Thank you very much for this insightful comment. To study the entry of iNVs into target cells, B16F10 were labeled with DiD and then incubated with DiO-labeled iNVs. It can be found in **Figure S20 that the iNVs (green fluorescence) were in the cytoplasm rather than fused with the target cell membranes (red fluorescence)**, demonstrating that iNVs enter the target cells via endocytosis rather than plasma membrane fusion.

Supplementary Figure S20. Immunofluorescence imaging of the entry of iNVs into B16F10 cells. Scale bar, 10 μm . Before the imaging, the cell nucleus, cell membrane and iNVs were labeled with DAPI (blue), DiD (red) and DiO (green), respectively.

Reviewer #2 (Remarks to the Author): expertise in STING agonist in cancer immunotherapy

Designer cell membrane nanovesicles amplify macrophage immune responses against cancer recurrence and metastasis

Here, Lang et al. designed novel nanovesicles (NVs), made up from the fusion of platelet-derived, M1 macrophage-derived and genetically-engineered (overexpressing high affinity CD47 receptor) cancer cell-derived extracellular vesicles and utilized them for cancer immunotherapy upon comprehensive characterization. Importantly, fusion vesicles (or so-called iNVs) showed potent anti-tumor effect by abrogating cancer recurrence and metastasis in murine tumor models of melanoma and to a lesser extent in triple negative breast cancer model, for which incorporation of an adjuvant, like cGAMP, into the NVs was necessary in order to get the optimal anti-tumor effect due to low immunogenicity of this tumor model. Mechanisms regulating the anti-tumor effect of iNV immunotherapy involved M1 polarization of TAMs, enhanced macrophage phagocytosis due to blockade of CD47-SIRP α signaling on TAMs and subsequent T cell recruitment to tumors. Therefore, such specially-designed NVs may offer therapeutic applications for cancer.

Response: Thank you very much for your positive and constructive comments. We have carefully revised the manuscript according to your comments and marked all the changes in red. Point-by-point responses to your comments are listed below.

Reviewer Comments:

1- As the authors also discussed, application of the anti-tumor NVs to the clinic is difficult, especially to produce clinical grade/GLP lot NVs. In the discussion, authors suggested that patient-derived NVs could be used to produce such anti-tumor NVs. However, have the authors examined the possibility that cancer patient-derived NVs may contain suppressive factors that can potentially halt anti-tumor immunity. Do they have any suggestion for this issue?

Response: Thank you very much for this insightful comment.

We agree that the iNVs may contain suppressive factors that potentially halt antitumor immunity. However, **we have confirmed that C-NVs, especially S α V-C-NVs, have encouraging antitumor performance (Figure S13), demonstrating that the advantages of stimulation factors on C-NVs are more than the disadvantages of suppressive factors.** It is worth mentioning that clinically applicable dendritic cell (DC) or DC-derived vesicle vaccine also contain suppressive factors, which could alleviate the concern on suppressive factors.

According to your comment, we also discussed this issue into the last part of the revised manuscript: “While the iNVs showed encouraging antitumor performance, the iNVs may contain suppressive factors that potentially halt antitumor immunity. **Gene knockout of certain suppressive factors and combination with immune checkpoint inhibitors may further improve the antitumor effects of iNVs.**”

Supplementary Figure S13. c, Tumor growth kinetics in different groups. d, Survival corresponding to the tumor size of mice after different treatments as indicated.

2- Starting from figure 3, authors compare therapeutic approaches using different types of NVs or anti-CD47 antibody treatment in murine tumor models. However, they compare 50 μ g of P-NV, M1-NV, anti-CD47 antibodies with 200 μ g of SaV-C-NV and 300 μ g of iNVs. Why do they use different doses for each agent rather than directly comparing the same dose of each NVs at least? Can the authors explain rationale behind this strategy?

Response: Thank you very much for this insightful comment.

In this work, the dose of 50 μ g of anti-CD47 antibodies was based on previous reports regarding CD47 immune blockade. Then, based on the B16F10 mouse model, we compared the *in vivo* effects of different concentrations of SaV-C-NVs and CD47 mAbs on the inhibition of tumor growth and found that **200 μ g of SaV-C-NVs showed similar antitumor effect as 50 μ g of anti-CD47 antibody (Figure S11)**. Thus, we employed 200 μ g of SaV-C-NVs in this work. Meanwhile, **we also tested the effect of M1-NVs concentration on macrophage polarization and used 50 μ g of M1-NVs (Figure S9)**. Finally, in order to obtain the best antitumor performance, we selected 50 μ g of P-NV, M1-NV, anti-CD47 antibodies with 200 μ g of SaV-C-NV and 300 μ g of iNVs to test the antitumor effects.

In the revised work, **we also normalized the dose of each NVs and repeated the *in vivo* experiments based on B16F10 tumor model**. Similarly, the iNVs showed the best antitumor effect on B16F10 tumor model, which was even better than cocktail therapy strategy (Figure S12). More discussion and experiment details can be found in the revised manuscript and supplementary information.

Supplementary Figure S12. c, Tumor growth kinetics in different groups. d, Survival corresponding to the tumor size of mice after different treatments as indicated.

3- Authors obtain M1-NVs from LPS-stimulated BMDMs raising a potential LPS contamination issue in M1-NVs, which are systemically injected to animals. Have the authors measured serum cytokines, such as IL-6, TNF α or type I IFNs, after i.v. injection of these NVs or observed any adverse effects related to this issue?

Response: Thank you very much for this careful and important comment.

Per your suggestion, **we tested the serum cytokine levels after the iNVs treatment and observed no significant differences in TNF- α and IL-6 (Figure S16)**, suggesting that the biosafety of iNVs.

In addition, End-Point Chromogenic Endotoxin Test Kit (BioEndo, China) was used to measure the residual LPS in the NVs. As shown in Figure S2, the residual LPS in the iNVs was not detectable. The limit of detection of this LPS Kit is 0.01 – 0.1 EU/mL, thus there are less than 0.01 EU/mL LPS remaining in the iNVs, meeting the requirements of in vivo experiments regarding LPS. Sincerely hope our revision could meet with your approval.

Supplementary Figure S16. Serum cytokine measurement. **a**, TNF- α : tumor necrosis factor- α . **b**, IL-6: interleukin-6.

Supplementary Figure S2. Measurement of LPS contents in M0-NVs, M1-NVs and iNVs.

4- NV treatment can enhance infiltration of not only CD8 but also CD4 T cells as observed in FACS plots in figures 2k and 4b into the TME. Have the authors examined those CD4 T cells? Are they Tregs or anti-tumor T cells? In other words, is the NV-mediated anti-tumor effect dependent on CD8 or CD4 T cells?

Response: Thank you very much for this important comment. Per your suggestion, in the revised work, **we used depleting antibodies against CD4⁺ or CD8⁺ T cells and demonstrated that the antitumor effects of iNVs were dependent on both of CD4⁺ and CD8⁺ T cells (Figure S17)**. More

discussion and experiment details can be found in the revised manuscript and supplementary information.

Supplementary Figure S17. c, Tumor growth kinetics in different groups.

5- In this study, authors designed iNVs by using the same type of tumor cells for each tumor model (e.g. B16 F10 NVs for melanoma model and 4T1 NVs for breast cancer model). Have the authors tried the vice versa, for example, using B16 F10 NVs in breast cancer model in order to show the tumor antigen-specificity of this anti-tumor effect? Another way to prove this would be utilization of a tumor cell line expressing a model antigen, such as OVA, and measure antigen-specific T (or even B) cell responses after ex vivo splenocyte re-stimulation (or measuring OVA-specific serum antibodies).

Response: Many thanks for this insightful comment. Per your suggestion, **we tested the antitumor effects of 4T1-NVs and 4T1-SαV-NVs on B16F10 tumor model.** It can be found in **Figure S13, B16-NVs and B16-SαV-NVs showed better antitumor effects on B16F10 tumor model than 4T1-NVs and 4T1-SαV-NVs, respectively,** which was presumably a result of tumor antigen specificity. It is worth mentioning that autologous cellular vesicles have been applied in cancer nanovaccine development due to the tumor antigen specificity (Nano Lett. 2014, 14, 2181-2188; Sci. Transl. Med. 2019, 11, eaat5690). More discussion and experiment details can be found in the revised manuscript and supplementary information. PS: we didn't test the antitumor effects of B16-SαV-NVs on 4T1 tumor model because iNVs only showed mild effect on 4T1 model (Figure 6).

Supplementary Figure S13. c, Tumor growth kinetics in different groups. d, Survival corresponding to the tumor size of mice after different treatments as indicated.

Reviewer #3 (Remarks to the Author): expertise in membrane nanovesicles

In this article, Rao et al. fused engineered cancer cell nanovesicles, platelet nanovesicles, and M1 macrophage nanovesicles into a single platform for the prevention of tumor recurrence and metastasis in a surgical setting. The iNVs had clear efficacy in surgical models and is versatile enough for drugs to be easily incorporated inside. While the benefits of the platelet and macrophage component was clearly shown, the need and advantages for the cancer cell nanovesicle component was lacking. It is unclear as to why the authors decided to study the benefits of each individual component without a comparison to their actual formulation. Moreover, the evidence for the nanovesicle fusion is limited. Based on the results provided, it is unclear if the authors actually fused the three distinct components together or simply administered them as a cocktail formulation. Below are some detailed comments for the authors to further improve the manuscript.

Response: Thank you very much for your meaningful and careful comments. According to your comments, we tried our best to revise the manuscript. More than 100 mice were involved in the revision to include more experimental controls. We also tried to improve the nanovesicle characterization in composition and mechanism of action, and to address the concerns related to cytotoxicity and animal number. We marked all the changes in red and point-by-point responses to your comments are listed below.

1. While the roles of P-NVs and M1-NVs were clearly defined, the necessity for engineered C-NVs is unclear. Can the author elaborate on this design?

Response: Many thanks for this constructive comment.

We used S α V-C-NVs because S α V-C-NVs could efficiently block the CD47-SIRP α signaling axis and showed synergistic effect with P-NVs and M1-NVs on the activation of macrophage immune responses against cancer recurrence and metastasis (Figure 2-4).

B16F10 and 4T1 cancer cells rather than platelets or macrophages were used for genetic over-expression of SIRP α due to the following two reasons: **1) compared with platelets and BMDMs, cancer cell lines are easier and more suitable for gene editing; 2) cancer cell NVs could provide specific tumor antigen, enhancing the antitumor effects (Figure S13, please see more details in Response #5 for Reviewer #2).**

2. Throughout the manuscript, the authors referred to M0-NVs several times, but did not explain what they are. Are these NVs derived from unstimulated BMDMs? Please explain.

Response: Yes, M0-NVs were derived from unstimulated BMDMs. According to your suggestion, we defined the M0 macrophages and M0-NVs: “when compared with **nonpolarized M0 macrophages** and **the derived M0-NVs**, respectively”.

3. For the fusion membrane, the author chose a protein ratio of 1:1:4 for P-NVs:M1-NVs:SαV-C-NVs. Can the authors please explain and provide data as to how this ratio was determined? Why was it chosen as opposed to a simple 1:1:1?

Response: Thank you very much for this insightful comment. In this work, the dose of 50 μg of anti-CD47 antibody was based on previous reports regarding CD47 immune blockade. Then, based on the B16F10 mouse model, we compared the *in vivo* effects of different concentrations of SαV-C-NVs and CD47 mAbs on the inhibition of tumor growth and found that **200 μg of SαV-C-NVs showed similar antitumor effect as 50 μg of anti-CD47 antibody (Figure S11)**. Thus, we employed 200 μg of SαV-C-NVs in this work. Meanwhile, **we also tested the effect of M1-NVs concentration on macrophage polarization and used 50 μg of M1-NVs (Figure S9)**. Thus, we chose a protein ratio of 1:1:4 for P-NVs:M1-NVs:SαV-C-NVs, rather than a simple 1:1:1.

4. While the fluorescence images in Figure 1h appears to show that there are some fusion NVs due to colocalization, the data itself is insufficient. The supporting WB analysis is a flawed design since there's no purification step involved in the fusion process. One can in theory get the same WB results if they just simply mix the three components at the same ratio. Please provide more supporting evidence that the final NVs are indeed a fusion of the three, particularly if the authors can show that it's fused at a 1:1:4 manner.

Response: Thank you very much for this important and insightful comment.

In our previous works (Adv. Funct. Mater. 2018, 28, 1803531; Adv. Funct. Mater. 2019, 29, 1807733), we tried to design specific immune magnetic beads (IMBs) and use pull down assay to verify the membrane fusion and purify the hybrid vesicles. **We demonstrated that the success rate of membrane fusion was almost 100% by sonication and extrusion.**

According to your suggestion, anti-CD14 antibody-modified IMBs were prepared (Figure S4a; CD14 is a specific protein for macrophages) and pull-down assay was further used to purify the iNVs. **We found that almost all the proteins were in the sediment and there was virtually NO protein in the supernatant (Figure S4b), demonstrating that P-NVs and SαV-C-NVs were completely fused with M1-NVs and pulled down by the IMBs.**

Supplementary Figure S4. (a) The design of the IMBs for the pull down assay. (b) Measurement of the protein content in the sediments and supernatants of iNVs after the pull down assay.

5. Cytotoxicity with a cancer cell line in Figure S4 is not a good representation of biocompatibility. The authors should repeat the experiment with a cell line of non-cancerous origin or with primary cells.

Response: According to your suggestion, we repeated the cytotoxicity experiment with a cell line of non-cancerous origin (human embryonic kidney 293T cell line) and demonstrated that iNVs possess good biocompatibility (Figure S6).

Supplementary Figure S6. (a) Human embryonic kidney 293T cells viability after incubation with P-NVs, M1-NVs, SαV-C-NVs and iNVs for and 48 h.

6. The schematic in Figure 2d was quite confusing. From the manuscript, it seems like these were two separate experiments. Can the authors revise it and try to make it clearer?

Response: According to your suggestion, we revised the schematic (please see below).

7. The mechanism of M1-M2 polarization is unclear. While there has been studies where extracellular vesicles can induce M2 polarization, that interaction is largely due to mRNAs/miRNAs that are naturally inside those vesicles. In this platform, those compounds should not exist since the cells were lysed and then the NVs are reformed. What is actually allowing M1-NVs to polarize M2 macrophages into the M1 phenotype?

Response: Thanks for this careful comment. **The M1-M2 repolarization by M1-NVs was dependent on the mRNA in the NVs. While most of the intracellular content was removed by lysis and reconstruction, there are residual mRNA in the M1-NVs (Figure 1e).** PS: in order to efficiently remove the intracellular content, DNase and RNase were used in the preparation of SαV-C-NVs.

Figure1. e, Relative mRNA expression of *Cd86*, *Tnf*, *Il6*, and *Inos* in M0-type macrophages (M0 cells), M1 cells, M0-NVs and M1-NVs.

8. The distinct dosages the authors chose for their therapeutic studies are quite odd, which can lead to an unfair comparison between the different groups. In the current scheme, the advantage of the fusion iNVs is abrogated. Arguably, the same therapeutic efficacy can be achieved with a cocktail administration. The authors should normalize the dosage to the same protein content in order for a fairer comparison.

Response: Thank you very much for this insightful comment.

As replied in Response #3, the dosages of different NVs involved in this work were based on in vivo animal experiments.

And per your suggestion, **we normalized the dose of each NVs and repeated the in vivo experiments based on B16F10 tumor model.** The iNVs showed the best antitumor effect on B16F10 tumor models, **which was better than the cocktail therapy strategy (Figure S12).**

Although SαV-C-NVs and M1-NVs could block the CD47-SIRPα signaling axis and promote the M2-M1 macrophage repolarization, respectively, they can't efficiently accumulate in the surgical sites and interact with CTCs in the blood circulation. **By fusing with P-NVs, the iNVs help the SαV-C-NVs and M1-NVs efficient accumulate in the surgical sites and interact with CTCs, thus enhancing the antitumor effects.**

Supplementary Figure S12. c, Tumor growth kinetics in different groups. **d,** Survival corresponding to the tumor size of mice after different treatments as indicated.

9. In Figure 3c, why does the iNVs group have 6 mice whereas all the other groups have 5 mice? This seems like an unfair comparison. Similarly, this is observed in Figure 6c for iNVs + cGAMP and iNVs@cGAMP groups. The authors should keep consistently the same number of mice in each group for their studies.

Response: Thanks for this careful comment. **While we normalize the animal number before the experiments, we usually add one or two more mice into the key treatment groups.** Similar situations can be found in other biomedical research (Nature 2020, 579, 260-264; Sci. Transl. Med. 2018, 10, ean3682).

REVIEWERS' COMMENTS:

Reviewer #1 (Remarks to the Author):

I am satisfied with the revisions. I recommend acceptance.

Reviewer #2 (Remarks to the Author):

The authors responded to the reviewers' comments properly.

Reviewer #3 (Remarks to the Author):

The authors have done a wonderful revision, in which all unclear points are now clarified and new experimental data is included to address technical question. I have no further comments.